# Harnessing the Endogenous Plasticity of Pancreatic Islets: A Feasible Regenerative Medicine Therapy for Diabetes?

**DOI:** 10.3390/ijms22084239

**Published:** 2021-04-19

**Authors:** Petra I. Lorenzo, Nadia Cobo-Vuilleumier, Eugenia Martín-Vázquez, Livia López-Noriega, Benoit R. Gauthier

**Affiliations:** 1Andalusian Center for Molecular Biology and Regenerative Medicine-CABIMER, Junta de Andalucía-University of Pablo de Olavide-University of Seville-CSIC, 41092 Seville, Spain; nadia.cobo@cabimer.es (N.C.-V.); eugenia.martinvazquez@cabimer.es (E.M.-V.); livia.lopez@cabimer.es (L.L.-N.); 2Centro de Investigación Biomédica en Red de Diabetes y Enfermedades Metabólicas Asociadas (CIBERDEM), 028029 Madrid, Spain

**Keywords:** diabetes, regeneration, β-cell heterogeneity, transdifferentiation, redifferentiation, single-cell transcriptomics, PAX4, LRH-1/NR52A, HMG20A

## Abstract

Diabetes is a chronic metabolic disease caused by an absolute or relative deficiency in functional pancreatic β-cells that leads to defective control of blood glucose. Current treatments for diabetes, despite their great beneficial effects on clinical symptoms, are not curative treatments, leading to a chronic dependence on insulin throughout life that does not prevent the secondary complications associated with diabetes. The overwhelming increase in DM incidence has led to a search for novel antidiabetic therapies aiming at the regeneration of the lost functional β-cells to allow the re-establishment of the endogenous glucose homeostasis. Here we review several aspects that must be considered for the development of novel and successful regenerative therapies for diabetes: first, the need to maintain the heterogeneity of islet β-cells with several subpopulations of β-cells characterized by different transcriptomic profiles correlating with differences in functionality and in resistance/behavior under stress conditions; second, the existence of an intrinsic islet plasticity that allows stimulus-mediated transcriptome alterations that trigger the transdifferentiation of islet non-β-cells into β-cells; and finally, the possibility of using agents that promote a fully functional/mature β-cell phenotype to reduce and reverse the process of dedifferentiation of β-cells during diabetes.

## 1. Introduction

Diabetes, a metabolic disease characterized by chronic hyperglycemia due to insufficient production of insulin to meet the insulin demand, is nowadays the most prevalent chronic disease affecting approximately 463 million adults worldwide, and its incidence is expected to rise up to 700 million people by 2045 (IDF, www.idf.org (accessed on 18 April 2021)). Diabetes development results from both polygenic and environmental contributors that lead to a progressive decrease in the functional β-cell mass, either due to autoimmune destruction of β-cells in type 1 diabetes (T1D) or due to β-cell dysfunction/dedifferentiation in type 2 diabetes (T2D) and gestational diabetes (GD) [1]. In both T1D and T2D, the disease progresses through an asymptomatic phase before being detected, since only after a reduction of 65–80% of the β-cell mass is when hyperglycemia develops [2,3]. Thus, the pathogenesis of diabetes is a continuum that can be divided into different stages associated with changes in β-cell mass, phenotype, and function (Figure 1) [4,5]. Current treatments for diabetes are based on the administration of exogenous insulin or insulin analogs, or treatment with agents that stimulate insulin secretion by β-cells, improve insulin sensitivity, promote glucose excretion, or delay carbohydrate digestion. However, despite their beneficial effects, these therapies do not prevent the development of life-threatening complications. The definitive therapy for diabetes, independently of the disease etiology, will rely on the restoration of the β-cell mass and function, allowing the recovery of the endogenous control of glucose homeostasis. Therefore, understanding processes affecting the functionality of the β-cell mass, together with the characterization of islet adaptive mechanisms, will open new avenues for the identification of “druggable” targets that can protect/regenerate endogenous β-cell, rendering a better control of blood glucose levels.

### The Pancreatic Islet

The islets of Langerhans are highly specialized endocrine micro-organs of the pancreas responsible for the control of body glucose levels. The islet is an amalgam of five different types of cells, glucagon-producing α-cells, insulin-producing β-cells, somatostatin-producing δ-cells, pancreatic polypeptide-producing PP-cells, and ghrelin-producing ε-cells, which are derived from a common NGN3^+^ endocrine progenitor (refer to [6] for a detailed review of islet organogenesis). At birth, pancreatic islets are still immature and defective in glucose-stimulated insulin secretion (GSIS) [7,8]. During early postnatal life, the pancreatic islets undergo a period of reorganization/proliferation and maturation to form fully functional islets, with multicellular interactions that ensure the correct blood glucose homeostasis. Different age-dependent gene expression programs are responsible for controlling the functional maturation of the islets. Juvenile-enriched gene sets are mainly associated with the unfolded protein response (UPR) and cell proliferation, while adult islets show an enrichment in genes associated with hormone production/secretion and cell signaling [9]. Remarkably, these changes in gene expression are associated with alterations in the histone modification profile. ChIP-seq analyses of the adult gene set associated with the physiological function of the islet, in both adult and juvenile islets, revealed that these genes are enriched in active H3k4me3 and H3K27ac marks in adult islets but also enriched in repressive H3K27me3 marks in juvenile islets, highlighting the importance of epigenetic regulation in the maturation of pancreatic islets [8,9]. After this maturation period, adult islets have been considered postmitotic, relatively quiescent organs, with limited capacity to adapt to environmental insults. However, this dogma has been challenged. Different studies have demonstrated that adult pancreatic islets possess an intrinsic plasticity, based, at least in part, on the heterogeneity detected in the insulin-producing β-cell population [10,11], that renders some of these subpopulations more resistant to environmental insults and prone to proliferation [12]. Likewise, under specific circumstances, islet non-β-cells can transdifferentiate into β-cells, contributing to the regeneration of the functional β-cell mass [13,14], and the regenerative capacity of the β-cells, which can be activated after the relief of the metabolic stress. Since diabetes is characterized by the inability of pancreatic β-cells to meet the insulin demand, due to either a nearly complete loss (T1D) or a deficit (T2D and GD) in functional β-cells, this intrinsic plasticity of adult islets opens the possibility of the development of novel regenerative therapies for the cure of this currently chronic disease. 

## 2. Heterogeneity within β-Cell Population of the Islets Can Modulate Islet Plasticity 

### 2.1. β-Cell Heterogeneity in Adult Islets

Evidence of β-cell heterogeneity among different islets and within islets was already reported in the 1960s (reviewed in [15]). The technical advances that have taken place since then, allowing in vivo cell labeling and high-throughput screenings, have confirmed these observations. Gene expression analysis of sorted β-cells from adult mice islets revealed the coexistence of several subpopulations within the β-cells [16]. Further studies using a reporter mouse model expressing GFP under the control of an insulin promoter confirmed this heterogeneity, showing the existence of three different subpopulations of β-cells based on insulin/GFP expression levels and cell granularity [17]. Furthermore, independent studies performed in both mouse and human islets, tracing the activity of the *Pdx1* and the insulin promoters using a dual reporter lentiviral system (expressing RFP under the control of the *Pdx1* promoter and GFP under an insulin promoter), disclosed the existence of two phenotypically different β-cell subpopulations based on their insulin expression levels: a predominant INS^HIGH^ population with increased expression in mature β-cell markers, such as *MafA*, *Nkx6.1*, *Slc2a2 (Glut2)*, and *Gck*, and an INS^LOW^ population that accounts for 15–20% of β-cells, with increased proliferation rate and increased expression of *Pax4, MafB*, and *Nkx2.2,* among other genes [18]. These studies clearly indicate the coexistence of several subpopulations of β-cells with different maturation statuses within adult islets [17,18,19]. Additional studies focused on the expression of the Wnt/polar planar effector Flattop (FLTP), expressed in terminally differentiated β-cells, revealed two subpopulations of β-cells: a mature FLTP^+^ population of metabolically active β-cells and a population of immature β-cells lacking FLTP that are responsible for the compensatory proliferation of β-cells under situations of increased insulin demand [20]. Further identification of aging markers for β-cells, such as p16^ink4a^, p53BP1, or IGF1R, reinforced the coexistence of mature and more immature β-cells within the same islet [11]. Importantly, these β-cell subpopulations are transient and dynamic, being able to evolve from an immature subpopulation to a more mature one [18,20], suggesting that these immature β-cells are a possible reservoir for mature high secretory β-cells. It is noteworthy that recent studies have identified in the periphery of the islets a subpopulation of functionally immature β-cells characterized by the lack UCN3 (urocortin 3) that persist throughout life [21]. Interestingly, these β-cells, termed “virgin β-cells,” are not originated from pre-existing β-cells but by the transdifferentiation of ⍺-cells [21,22]. 

The coexistence within an islet of more immature and proliferative β-cells together with more mature quiescent secretory β-cells confers a functional flexibility to the islet, allowing the expansion of the β-cell mass when needed, without a detrimental effect on the endocrine function. Remarkably, the relative proportions of these subpopulations vary with aging and also under metabolic stress conditions, accumulating the aged β-cells concomitantly with a reduced proliferative capacity of the islets [11,17]. These described age-related changes in the proportions of β-cell subpopulations may stem from the loss of β-cell plasticity in aging, correlating with increased T2D incidence, with the majority of patients being above 50 years of age (https://idf.org/ (accessed on 18 April 2021)). The heterogeneity within the islet β-cells and its regulation under stress conditions as well as in aging reveals its importance for adequate islet functionality and imposes a redefinition in the design of novel therapeutic approaches to stimulate β-cell expansion, which will have to maintain the β-cell heterogeneity. This will require a better characterization of these subpopulations of β-cells, their physiological role, and the regulatory mechanisms that control their proportions. 

### 2.2. Functional Implications of β-Cell Heterogeneity 

The development of antibodies against cell surface markers of β-cells revealed the existence of four antigenically distinct subpopulations of β-cells in human islets that, despite an overall transcriptomic similarity, showed differential regulation for specific gene sets, among them, genes associated with insulin secretion and genes known to be dysregulated in T2D [10]. Further characterization of these subpopulations of β-cells showed functional differences among them, in both basal and stimulated insulin secretion [10]. A recent study combining whole-cell patch-clamp measurements and single-cell RNA sequencing (scRNA-seq) demonstrated heterogeneity in excitability among the different β-cell subpopulations characterized by their transcriptomic profile [23]. One could think that these differences in functionality may be the result of differences in the maturation status of the cells, with poorly functional immature β-cells. Notwithstanding, recent studies have revealed a remarkable implication of immature β-cells in the coordination of insulin secretion. Analysis of Ca^2+^ dynamics in the islet showed that after glucose stimulation, β-cells form a network that allows the synchronous propagation of Ca^2+^ waves. These Ca^2+^ waves are fired by rare superconnected cells, termed hubs, whose activity precedes and outlasts the activity of the rest of β-cells, functioning like a pacemaker [24]. Specific destruction of these hubs has catastrophic consequences for coordinated islet response to high glucose, revealing their implication in the control and coordination of glucose-dependent insulin release [24,25]. Further characterization of these hubs demonstrates that these cells are highly metabolic and transcriptionally immature GCK^HIGH^/PDX1^LOW^ β-cells [25]. Reinforcing the role of immature β-cells in coordinating insulin secretion, ectopic expression of PDX1 and MAFA in isolated islets, and therefore increasing the proportion of mature PDX1^HIGH^/MAFA^HIGH^ β-cells in the islet, caused a reduction in Ca^2+^ responses to glucose, concomitantly with a decrease in the proportion of hubs in these islets [26]. Interestingly, functional differences among the mature INS^HIGH^ subpopulations of β-cells have also been described. A recent study using single-molecule fluorescence in situ hybridization (smFISH) that allows the visualization of individual mRNA molecules identified a subpopulation of β-cells that contain twofold higher *Ins* mRNA than the median expression in the islet, denoting these cells as “extreme” β-cells [27]. Interestingly, these “extreme” β-cells, despite their higher expression levels of *Ins* mRNA, have significantly lower insulin protein levels and are relatively depleted of insulin granules when compared with other INS^HIGH^ β-cells, which has suggested their possible specialization in basal insulin secretion [27]. Remarkably, the abundance of these “extreme” β-cells is increased in the db/db mouse model for insulin resistance that lacks first-phase insulin secretion in response to glucose, also suggesting an association between the lack of first-phase insulin secretion and the change in the proportion of “extreme” β-cells [27]. Altogether, these functional studies clearly indicate the requirement of the coexistence of different β-cell subpopulations to integrate and coordinate an adequate islet response, revealing an important role for the INS^LOW^ more-immature β-cells, besides being a reservoir for mature β-cells.

### 2.3. Maintenance of β-Cell Heterogeneity 

These described specific roles for some of the β-cell subpopulations suggest that the heterogeneity within the β-cells may not be solely the result of different β-cells’ age, but rather by the involvement of molecular mechanisms actively controlling this heterogeneity. In agreement with this, scRNA-seq and pseudotemporal ordering of human β-cells suggest that β-cells transition between periods of activity, associated with higher expression of insulin and lower expression of UPR-related genes, and periods of rest/recovery from stress that are characterized by lower expression of insulin and higher expression of UPR-related genes [28]. Thus, at any time point, a subpopulation of β-cells would be responsible for fulfilling the insulin requirements, while other β-cells would be recovering from the stress of their biosynthetic activity. It is noteworthy that previous studies in mouse have revealed that activation of UPR response stimulates β-cell proliferation [29]. Therefore, the entrance of the β-cells into a recovery state characterized by INS^LOW^UPR^HIGH^ may also activate pro-proliferative mechanisms that render these β-cells prone to proliferation. This is an important issue for the development of novel regenerative approaches since it opens the possibility to specifically stimulate the proliferation of the “resting” cells, while keeping functional the “active” cells responsible for insulin secretion. Further studies are required to demonstrate the feasibility of this approach, which will probably be based on noncontinuous treatments, ensuring the maintenance of both subpopulations and the transitions between them.

As mentioned before, the different β-cell subpopulations do not respond in the same way to stress conditions. While the hubs are metabolically fragile cells and preferentially targeted by the environmental insults [24], other populations are more resistant to environmental stressors. Therefore, exposure of the islets to a metabolic stress that causes alterations in the proportions of these subpopulations [11] can affect the overall functionality of the islet. In agreement with this, studies using scRNA-seq analysis of sorted β-cells in the STZ mouse model of diabetes revealed the existence of three different subpopulations with different sensitivities to STZ-induced cell death [30]. These β-cell subpopulations are not equally functional, and the lack of the STZ-sensitive subpopulation renders the islets unable to control glucose homeostasis. In agreement with this pathogenic alteration in β-cell subpopulations, the four antigenically distinct β-cell subpopulations identified by Dorrell and colleagues showed similar proportions among healthy donors, but were clearly different in islets from T2D donors [10]. Altogether, these data strengthen the hypothesis that β-cell heterogeneity is a dynamic and regulated process with an important role in islet functionality and thus in glucose homeostasis (Figure 2). Therefore, successful strategies for regenerating a functional β-cell mass in diabetes should maintain this heterogeneity of the β-cells within the islet.

### 2.4. PAX4 in β-Cell Heterogeneity 

Understanding the origin/maintenance of this heterogeneity in β-cells will be crucial for the development of novel regenerative approaches. This will require full characterization of the transcriptome profile of these subpopulations and identification of the molecular mechanisms involved in the maintenance of β-cell heterogeneity. One of the examples of this heterogeneity within islet β-cells stems from the expression of PAX4, a pro-proliferative and antiapoptotic transcription factor that is essential for β-cell generation [31,32,33,34,35]. Remarkably, mutations/polymorphisms in *PAX4* have been associated with increased diabetes risk in humans (review in [36,37]). During embryonic development, PAX4 is expressed in all endocrine progenitors and is mandatory for their commitment to and the further maturation of β-cells [12,38,39]. During postnatal life, *Pax4* expression decreases with age, becoming restricted to a subpopulation of β-cells that accounts for 30% of the β-cells in adult mouse islets and 15% of the β-cells in aged mice [12]. During adulthood, PAX4^+^ β-cell population is dynamic and can transiently expand during situations of increased insulin demand, such as pregnancy, correlating with the increase in β-cell proliferation. It is noteworthy that the analysis of the proliferating β-cells during pregnancy showed a preferential expansion of the PAX4^+^ subpopulation [12]. Remarkably, *Pax4* expression levels also increase in the STZ mouse model of diabetes [40], and in vivo ectopic overexpression of this factor enhances β-cell resistance to STZ [35], as well as against an autoimmune attack in a mouse model of experimental autoimmune diabetes [41]. As expected, based on these described actions of PAX4, the subpopulation of PAX4^+^ β-cells presents higher resistance/survival after in vivo STZ treatment, as well as after ER stress induction by thapsigargin treatment in isolated islets [12]. It is noteworthy that the β-cell subpopulation that escapes from STZ-induced cell death identified by Feng and colleagues showed low levels of *Slc2a2 (Glut2)* [30], and PAX4 represses *Glut2* expression [35], suggesting a possible overlap between these two subpopulations. Altogether, these data suggest that PAX4, a known master regulator of β-cell development, is also a key regulator for the maintenance of a more immature, more resistant, and proliferation-prone β-cell subpopulation, which may be targeted for the stimulation of the replenishment of the more mature β-cell subpopulations [37,42].

### 2.5. Future Directions in β-Cell Heterogeneity Studies 

One of the remaining questions is the identification of the mechanisms involved in the origin and maintenance of this heterogeneity within the β-cells. Studies in mouse models suggest that β-cell heterogeneity already arises during islet organogenesis [20]. Moreover, the cell–cell contacts established by β-cells within the islets and their exposure to the locally released factors also have a significant impact on β-cell maturation and insulin secretion [43,44], revealing the importance of the islet architecture by itself in the modulation of the heterogeneity of β-cells. However, further studies are needed to fully elucidate the mechanism involved in the generation/maintenance of the β-cell heterogeneity and its implication in the plasticity and functionality of the islets under physiological and pathophysiological conditions.

One of the actual limitations of these studies on β-cell heterogeneity is the lack of adequate markers for the identification of live β-cell subpopulations. The vast majority of studies performed in live β-cells have been done in mouse islets, taking advantage of mouse models for lineage tracing. However, to be able to extrapolate these data to human islets, aiming at the development of novel antidiabetic therapies, we need β-cell markers that are able to distinguish between the different β-cell subpopulations in live cells. The importance of live cell analysis in islet studies has prompted the search for novel surface markers that will allow the imaging and isolation of human pancreatic live β-cells. However, most of the reported cell surface markers are common to all β-cells [45,46], not being able to discern among the different subpopulations. To our knowledge, the only study focused on the identification of β-cell surface markers distinguishing the different β-cell subpopulations is the one by Dorrell and coworkers [10]. In this study, the authors demonstrated the feasibility of separating four different subpopulations among β-cells based on the expression of the cell surface markers ST8SIA1 and CD9. Nevertheless, it should be noted that neither ST8SIA1 nor CD9 is specific to β-cells. We are confident that the recent advances in scRNA-seq will facilitate the characterization of novel cell surface markers for the identification and isolation of live β-cell-specific subpopulations in both murine and human islets. 

In parallel, several studies have centered on the identification/development of dyes as an alternative approach for the characterization of live β-cell subsets. The differences in metabolic activity and/or electrophysiological response of β-cell subpopulations were exploited for the development of these novel dyes, which will allow the live imaging of β-cell subpopulations and their dynamics [47]. These important advances in the live imaging of β-cells will report important information on β-cell heterogeneity.

## 3. Transdifferentiation of Pancreatic α-Cells into β-Cells

### 3.1. ⍺-Cells as Source for New β-Cells

The lack of enough functional β-cells in diabetes has stimulated studies aiming at the generation of new β-cells to replenish β-cell mass during the progression of the disease. Based on islet development studies, together with the characterization of the β-cell transcriptome, numerous studies have shown the feasibility of direct reprogramming (transdifferentiation) of non-β adult cells into insulin-producing β-cells. The fact that all islet cells derive from a common NGN3^+^ early endocrine progenitor, which delaminates from the embryonic ductal tree to form clusters of endocrine cells that will then self-organize into pancreatic islets [48], indicates a close relation among the different cell types in the islet, suggesting an “easier” reprogramming among them. Additionally, the hyperplasia of ⍺-cells in diabetic islets [49], together with the fact that T2D is also associated with hyperglucagonemia that worsens the hyperglycemia in diabetes (review in [50]), suggests that the ⍺- to β-cell transdifferentiation would confer a dual benefit, regenerating β-cell mass and, at the same time, reducing the number of ⍺-cells. Thus, ⍺-cells are good candidates as a source for the generation of new β-cells. Supporting the possibility of ⍺- to β-cell transdifferentiation for diabetes treatment, in vitro studies in cell cultures have shown that overexpression of β-cell factors such as *PDX1*, *MAFA*, and *NKX6.1* can transform human adult ⍺-cells into insulin-secreting cells that are able to form pseudo-islets with nearly normal insulin secretion [51]. Additionally, molecular analysis of the open chromatin regions (ATAC-seq and FAIRE-seq) in α- and β-cells, combined with transcriptomic analysis and histone modification profiles, revealed that many of the β-cell marker genes are bivalently marked (H3K4me3/H3K27me3) and have an open chromatin structure in ⍺-cells, suggesting that these genes are poised for activation in ⍺-cells [52]. It is noteworthy that the ⍺-cell signature genes are only associated with an open chromatin status in ⍺-cells, suggesting that ⍺-to β-cell transdifferentiation will be favored over the reverse process. Altogether, these data pinpoint ⍺-cells as a promising source for β-cell generation in regenerative therapies for diabetes treatment.

It is noteworthy that recent scRNA-seq studies of mouse islets have shown the coexistence of distinct clusters of ⍺-cells, which show a different response to HF diet [53]. Moreover, in T2D human islets the appearance of an impaired ⍺-cell function phenotype is not homogeneous in all ⍺-cells, but largely restricted to a subgroup of cells [54]. Further studies using whole-cell patch clamp combined with scRNA-seq analysis indicates a heterogeneous function and transcriptome regulation in human ⍺-cells [23], which was confirmed by the existence of two subpopulations of ⍺-cells based on their high or low glucagon content [55]. In agreement with this heterogeneity in ⍺-cells, a considerable variability in the electrophysiological response among different ⍺-cells was previously reported [56]. It is tempting to speculate that this heterogeneity within ⍺-cells might render some of these populations more susceptible to transdifferentiation; however, to validate this hypothesis further, studies on ⍺-cells are required. 

### 3.2. ⍺- to β-Cell Transdifferentiation

During pancreatic development, the commitment of the endocrine precursor cells into an ⍺- or β-cell lineage is controlled by the expression of two mutually exclusive factors: ARX and PAX4. The critical role of these transcription factors was clearly demonstrated in their specific KO models. *Arx* depletion causes a lack of ⍺-cells with a concomitant increase in δ- and β-cells, while *Pax4* ablation results in the absence of β-cells with a gain in ⍺-cells [31,57]. In double-KO animals lacking both *Pax4* and *Arx*, δ-cell fate is favored at the expense of ⍺- and β-cell lineages, without significant alteration in the total endocrine cell content [33]. Moreover, ectopic expression of *Pax4* in ⍺-cells [58,59], as well as in δ-cells [60], or combined inhibition of *Arx* and *Dnmt1* in ⍺-cells [61] leads to in vivo transdifferentiation of these cells into β-like insulin-producing cells. These data clearly established the feasibility of in vivo reprogramming of ⍺-cells into β-like insulin-producing cells. More importantly, extreme β-cell ablation in adult mice triggers the transdifferentiation of ⍺-cells into β-cells [13,62] in the absence of any genetic manipulation, a demonstration of the intrinsic plasticity of the adult islets to restore the β-cell mass. Moreover, the existence at the periphery of murine islets of a neogenic niche of “virgin β-cells” derived from the transdifferentiation of α- to β-cells [21] clearly demonstrates the endogenous capacity of α-cells to transdifferentiate into β-cells. 

The transdifferentiation of α- to β-cells progresses through a bihormonal INS^+^/GCG^+^ state of the cells. These bihormonal INS^+^/GCG^+^ cells have been identified in both human and murine islets, and their abundance increases under diabetic conditions. Interestingly, α-cell-derived bihormonal INS^+^/GCG^+^ cells detected in T1D islets lack DNMT1 and ARX [61], suggesting that similar to murine islets, ⍺- to β-cell transdifferentiation can also be stimulated in human islets by DNMT1 and ARX depletion. The proof of concept of ⍺- to β-cell transdifferentiation in vivo has stimulated numerous studies aiming at potentiating this transdifferentiation process as a possible novel therapy for diabetes treatment. The fact that the bihormonal INS^+^/GCG^+^ cells are rare in normal islets, but increase their abundance under diabetic conditions in both human and murine islets, suggests that this endogenous transdifferentiation process is activated under stress conditions, and may be mediated, at least in part, by signals emitted by β-cells [63]. However, under chronic stress conditions, this regenerative process is not sufficient to maintain an adequate β-cell mass to control glucose homeostasis, eventually leading to diabetes. Therefore, great efforts have been made towards the stimulation of this endogenous transdifferentiation process aiming at the development of regenerative therapies for diabetes.

#### 3.2.1. GLP-1 Stimulates ⍺- to β-Cell Transdifferentiation

GLP-1 (glucagon-like peptide-1), a widely studied incretin for its antidiabetic effects, has been shown to stimulate islet cell transdifferentiation. GLP-1 is mainly secreted by L-cells in the gut after feeding, but can also be produced and secreted by a high percentage of α-cells [64]. However, GLP-1 is rapidly inactivated by the enzyme dipeptidyl peptidase 4 (DPP-4). Activation of GLP-1 signaling by in vivo administration of GLP-1, GLP-1 analogs resistant to DPP-4, or DPP-4 inhibitors can ameliorate hyperglycemia in murine models and stimulate ⍺-cell proliferation, as well as ⍺- to β-cell transdifferentiation [49,65,66,67]. Additionally, treatment with liraglutide, a GLP-1 analog, has been shown to stimulate mouse β-cell proliferation and β-cell neogenesis [66,67]. Besides the endocrine action of circulating GLP-1, the paracrine action of ⍺-cell GLP-1 has been shown to be important for β-cell functionality [64]. It is noteworthy that GLP-1 production by α-cells is stimulated by stress conditions, such as insulin resistance [64,68] and the loss of β-cells by themselves [66]. Remarkably, in islets from nondiabetic human donors, insulin resistance correlates with increased α-cell mass and intra-islet GLP-1 production, indicating the activation of GLP-1 production by α-cells prior to hyperglycemia development [69]. Studies using different mouse models of insulin resistance, including pregnancy, have revealed a significant activation of prohormone convertase 1/3 (PC1/3) in adult α-cells, which resulted in the processing of the proglucagon precursor into GLP-1, correlating with the expansion of the β-cell mass [70]. Thus, the paracrine action of α-cell GLP-1, stimulated by environmental stress, may be an adaptive response of the islets. The increased insulin demand can stimulate a crosstalk between ⍺- and β-cells, activating the proliferation of ⍺-cells and their production of GLP-1, which in turn stimulates insulin secretion by β-cells and the expression of β-cell signature genes, such as *Pdx1,* in ⍺-cells, which may promote their transdifferentiation into β-cells. Despite the need for further studies to understand the molecular mechanism action underlying GLP-1-induced ⍺- to β-cell transdifferentiation, the partial repression of this response in FGF21-KO mice suggested that this factor could be mediating, at least in part, this action through activation of transcription factors, such as PDX1 and NGN3 in ⍺-cells [49]. Additionally, inhibition of the PI3K/AKT/FOXO1 signaling pathway by a PI3K inhibitor-blocked GLP-1 mediated an increase in *Pdx1* and *MafA* in ⍺-cells [65]. It is noteworthy that despite the fact that GLP-1 treatment fails to stimulate human β-cell proliferation, we have seen that GLP-1 treatment in 15 mM glucose stimulates the expression of the β-cell plasticity gene *PAX4* in human islets [71]. This increase in *PAX4* expression could be indicative of both a possible ⍺- to β-cell transdifferentiation, since PAX4 determines the commitment to the β-cell lineage, and an expansion of the β-cell subpopulation primed to proliferation that we described in the previous section. However, further studies are required to fully understand this effect on human islets. 

#### 3.2.2. Can GABA Stimulate ⍺- to β-Cell Transdifferentiation?

Although controversial, another factor that has been related to ⍺- to β-cell transdifferentiation is GABA (γ-aminobutyric acid). GABA is an inhibitory neurotransmitter widely present in the nervous system but also produced/secreted by non-neuronal tissues, including the endocrine pancreas [72]. In the islet, GABA is synthesized and released by β-cells and is likely involved in tuning the islet responses to glucose [73]. In vivo, GABA treatment of mouse models of diabetes improves glucose homeostasis by stimulating β-cell proliferation and decreasing apoptosis [74]. Consistent with these results, long-term treatment of wild-type mice causes islet hypertrophy mainly due to the increase in β-cell numbers [75]. Remarkably, islets derived from wild-type mice treated with GABA display an increase in bihormonal INS^+^/GCG^+^ cells concomitantly with a decrease in ARX^+^ α-cells and misexpression of *Pax4* in α-cells, suggesting the involvement of α- to β-cell transdifferentiation in the observed increase in β-cell mass [75]. Further analysis of the effect of long-term GABA treatment using the STZ model of diabetes revealed that, under diabetic conditions, GABA treatment allows the recovery of normoglycemia by promoting ⍺- to β-cell transdifferentiation, which is blocked by an α-cell-specific GABA_A_ receptor antagonist [75]. Interestingly, these authors showed that GABA treatment can also induce an increase in β-cell mass at the expense of ⍺-cells in human islets transplanted in mice, suggesting that GABA-dependent transdifferentiation capacity is also present in human α-cells. In agreement with this, treatment of α-cells with artemisinins, which stimulate GABA signaling, triggers the cytoplasmic translocation of ARX and, therefore, its inhibition, with subsequent loss of α-cell identity and the induction of insulin expression [76]. These data agree with previously reported α- to β-cell transdifferentiation after *Arx* depletion [77]. This artemisinin-induced ⍺- to β-cell transdifferentiation was further confirmed in vivo in rats and zebrafish using lineage tracing experiments [76]. However, these promising results were challenged by two independent studies in isolated human islets [78] and in vivo in a transgenic mouse model that allows lineage tracing of α-cells [79]. It is noteworthy that this last study was performed in normoglycemic animals, unlike the former study of Ben-Othman and colleagues, which used the STZ model of diabetes, which could, at least partially, explain these discrepancies. Importantly, further collaborative studies among these groups have been proposed to solve this controversy [80].

#### 3.2.3. Immune-Coupled ⍺- to β-Cell Transdifferentiation: Role of LRH-1

In the case of autoimmune diabetes, modulation of the immune system from a proinflammatory and autoimmune destructive environment to an anti-inflammatory and tissue remodeling milieu has been shown to stimulate ⍺- to β-cell transdifferentiation. Agonistic activation of the ligand-dependent nuclear receptor LRH-1 (liver receptor homolog-1, also termed NR5A2) by the small molecule BL001 [81] in human islets in culture prevented cytokine- and STZ-induced cell death and improved survival and glucose-stimulated insulin secretion in T2D islets [82]. It is noteworthy that islets treated with BL001 showed a stimulation of glucocorticoids secretion [82], which has been shown to improve islet survival [83]. Long-term in vivo administration of BL001 in different murine models of T1D prevents the development of the disease. Interestingly, BL001 treatment was shown to promote β-cell regeneration in the absence of increased proliferation, a process that appears to proceed through ⍺- to β-cell transdifferentiation with the appearance of bihormonal INS+/GCG+ cells. Remarkably, in a mouse model of experimental autoimmune diabetes, BL001-mediated β-cell regeneration occurred concomitantly with the expansion of the anti-inflammatory Tregs and T helper 2, and primed macrophages towards the immunosuppressive and tissue remodeling M2 phenotype [82]. Interestingly, the abundance of the bihormonal INS+/GCG+ cells after BL001 treatment in this mouse model was markedly higher than in the STZ model, indicating the important contribution of Tregs and M2 macrophages in ⍺- to β-cell transdifferentiation [82,84]. Supporting this immune-coupled β-cell regeneration, the combined therapy of induction of MHC-mismatched mixed chimerism to attenuate the immune response together with gastrin and EGF administration resulted in β-cell regeneration and glucose homeostasis recovery in firmly established diabetic NOD mice [85]. Lineage-tracing studies revealed that ⍺- to β-cell transdifferentiation is the major source of β-cells, which only reaches these levels of activation after attenuation of the immune response [85]. This integrated non-mutually exclusive and mandatory islet-immune dialogue may provide some clues on failures of clinical trials targeting solely the immune system, which will likely compromise β-cell replenishment [84,86].

Altogether, these data clearly establish the feasibility of regenerating β-cell mass and recovering glucose homeostasis through the activation of ⍺- to β-cell transdifferentiation (Figure 3). However, this also reveals the importance of the diabetic milieu in activating this process. Both the metabolic stress imposed by the development of T2D and the attenuation of the autoimmune attack that triggers T1D are pivotal players allowing and/or stimulating the activation of ⍺- to β-cell transdifferentiation. In agreement with this, while in healthy mice self-replication of pre-existing β-cells is the major mechanism responsible for the maintenance of the β-cell mass [22,87], upon metabolic challenge, a significant contribution from non-β-cells was observed [88].

## 4. Redifferentiation of β-Cells 

### 4.1. Dedifferentiation of β-Cells during Diabetes

The progression of T2D has long been characterized by a significant reduction in β-cell mass, initially assumed to be due to increased cell death. However, further studies using lineage tracing in animal models revealed that one of the key events in the onset and progression of T2D is the dedifferentiation of β-cells, a process implicating a regression towards a more immature progenitor-like stage [89,90]. This loss of β-cell identity has been associated with the loss of FOXO1 function [89]. Substantiating the existence of this dedifferentiation process also in human islets, islets from a T2D donor showed reduced insulin content and secretion and impaired β-cell exocytosis [23]. Moreover, single-cell transcriptomic studies of human islets from nondiabetic donors of different ages and T2D donors confirmed that a substantial fraction of T2D β-cells revert to an immature expression profile characterized by derepression of the newborn gene set [91]. Interestingly, the promoters of these reactivated genes display a bivalent chromatin state in normal adult β-cells [91], suggesting the possible participation of epigenetic regulation. Analysis of human islets from T2D patients showed that dedifferentiation of β-cells, characterized by loss of identity, dysfunction, and decreased regeneration capacity, is also implicated in the progression of the human disease [92,93], and has been suggested to be the pathological basis of β-cell failure in T2D [94]. This “selfish” behavior of the β-cells may facilitate their survival under metabolic stress conditions, albeit to the detriment of insulin secretion. β-cell dedifferentiation and dysfunction under insulin-resistant conditions cause hyperglycemia, which would in turn promote further β-cell dedifferentiation, activating a vicious circle that will eventually lead to frank diabetes [94]. Despite being more studied in T2D [95], dedifferentiation of β-cell is not exclusively a T2D-related process, since dedifferentiated dysfunctional INS^LOW^ β-cells also appear during the progression of autoimmune diabetes both in NOD mice [85,96] and in T1D patients [97]. 

### 4.2. Redifferentiation of Dedifferentiatiated β-Cells: Mimicking Organogenesis 

In vitro expansion of human pancreatic islet preparations also induced dedifferentiation of β-cells towards a mesenchymal cell phenotype, but remarkably, these dedifferentiated β-cells retained the potential to be redifferentiated into insulin-producing β-cells [98,99]. This observation suggests the possibility of stimulating the redifferentiation of the dedifferentiated β-cells in diabetes (Figure 4). In agreement with this, prediabetes, a high-risk state for diabetes characterized by insulin resistance and decreased β-cell function, can be reverted to normoglycemia if the metabolic stress is reduced [100,101]. The decrease in insulin demand allows the β-cell to enter a resting state in which it can be redifferentiated into a fully functional β-cell [101]. Similarly, a therapy with basal insulin or with insulin sensitizer drugs, which decrease the load on the β-cells, can outperform insulin secretagogues as first-line treatment for T2D [93,102]. Taking advantage of the possibility of tracking these dedifferentiated β-cells in lineage-tracing experiments in mouse models of diabetes, Wang and colleagues demonstrated that insulin therapy stimulates in vivo the redifferentiation of the dedifferentiated β-cells into mature NGN3^−^/INS^+^ β-cells [90]. These studies highlighted the feasibility of stimulating β-cell redifferentiation in diabetes patients to recover a functional β-mass able to control glucose homeostasis as a promising therapeutic approach for diabetes treatment. Understanding the mechanisms that take place during islet maturation as well as for the maintenance of a differentiated state will be of high relevance to develop a successful therapy for redifferentiation.

Endocrine pancreas differentiation is regulated by dynamic changes in gene regulatory networks sequentially activating numerous pathways to induce commitment and differentiation into the various islet cell types. The described maturation heterogeneity of β-cells within the adult islets has revealed the importance of these pathways also in the adult islets. Numerous studies have identified many transcription factors involved in the regulation of β-cell development, differentiation, and function, whose sustained loss can cause β-cell dedifferentiation and diabetes (recently reviewed in [103]). Some of them need to be expressed during commitment to β-cell and need to be downregulated for the acquisition of a fully mature β-cell phenotype, such as PAX4 [37], while others, such as MAFA, are involved in the last steps of β-cell maturation and are required for adequate insulin secretion in response to glucose [104]. This hierarchical regulation converges in the expression of a set of lineage-defining transcription factors in mature β-cells, including PDX1, PAX6, NKX6.1, NKX2.2, FOXO1, and MAFA. Alterations on the expression of these transcription factors and their downstream effectors cause the loss of the β-cell identity and functionality, and ultimately diabetes. Deletion of *Pdx1* in adult β-cells leads to the loss of the expression of β-cell key genes and the expression of some ⍺-genes [105]. The loss of Nkx6.1 in adult β-cells leads to the loss of genes involved in insulin biosynthesis and secretion [106]. MafA has also been shown to be important in maintaining the transcriptional program of adult β-cells and to prevent the expression of disallowed genes [107]. Pax6 is also essential for the maintenance of β-cell identity, since loss of Pax6 leads to gradual loss of insulin expression and the induction of the expression of genes characteristic of other islet cell types [108]. Similarly, Nkx2.2 deletion causes the repression of β-cell genes and the upregulation of other islet cell-type genes [109]. Interestingly, many of these transcription factors (PDX1, PAX6, NKX6.1, and NKX2.2) have been shown to be regulated by long noncoding RNAs (lncRNAs) that are differentially expressed in animal models of diabetes or/and in pancreatic islets from T2D donors [110,111,112,113]. Modulating the expression of these lncRNAs to the same direction as in healthy individuals promotes the redifferentiation of β-cells. Therefore, modulation of key lncRNAs and subsequent alteration of β-cell transcription factors by antisense oligonucleotides, which are already in clinical trials for several cancers, can be a promising strategy for the treatment of T2D by preventing and/or reversing the loss of β-cell identity [114].

### 4.3. HMG20A Role in Functional Maturation of β-Cells 

Nevertheless, pancreatic β-cell identity is defined by not only the expression of the maturity genes but also by the selective repression of other non-β-cell programs [109] and of specific genes widely expressed elsewhere in the body, the disallowed genes [115]. It is noteworthy that these “disallowed genes” are upregulated under conditions of metabolic stress, such as high fat diet and T2D [95]. The repression of disallowed genes concomitantly with the activation of maturity genes important for insulin transcription, processing, packaging, and secretion reveals the existence of an on/off switch mechanism in β-cells, which coordinate the time frame expression of these genes during maturation. This resembles the well-studied regulation of neuronal maturation, where the expression of REST (RE silencing transcription factor, also known as NRSF, neuron-restrictive silencer factor) in neuronal precursors inhibits the expression of neuronal genes [116]. Neuronal precursor maturation correlates with the repression of REST, thus allowing the expression of neuronal genes. It is noteworthy that *REST* is a β-cell “disallowed gene,” ubiquitously expressed in most cells of the body with the exception of mature neurons and pancreatic β-cells, despite its expression in both neuronal precursors and embryonic β-cells [117]. Aberrant induction of REST expression in neurons or progenitors plays a role in neurodegenerative and neurodevelopmental diseases, while reactivation of REST in adult mouse β-cells leads to their dedifferentiation and hyperglycemia [117,118]. Remarkably, mature β-cells express a number of REST target genes, identified as neuronal traits, which are crucial for β-cell identity [119]. Interestingly, the chromatin remodeler factor HMG20A (also known as iBRAF), an inhibitor of the LSD1/CoREST complex, and initially identified in neurons, is one of the loci associated with both T2D and GDM [120,121]. HMG20A has been described to bind to the LSD1/CoREST complex, displacing HMG20B, and triggering the detachment of the complex from the DNA, thus alleviating the transcriptional repression imposed by the LSD1/CoREST complex [122]. It is noteworthy that decreased expression of *HMG20A* has been found in islets from T2D donors when compared with islets from normoglycemic donors [123], suggesting that low levels of HMG20A may result in the activation of the inhibitory action of the LSD1/CoREST complex, causing the dedifferentiation of β-cells. Supporting this hypothesis, our recent studies have shown that HMG20A is required for the functional maturation of the β-cells [123]. Silencing of HMG20A in isolated islets and in β-cell lines results in the upregulation of *Rest* and *Pax4* and the repression of *NeuroD*, *Gck*, *MafA* and *Ins* with concomitant blunt of GSIS. Interestingly, during gestation, one of the physiological conditions that trigger β-cell expansion, the expression of this chromatin remodeler, transiently increases in the last phase of gestation, correlating with the decrease in β-cell proliferation [123]. These data suggest that HMG20A is required for the functional maturation of the newly formed β-cells, by, at least in part, tuning down PAX4. Thus, activation of the pathways controlled by HMG20A could lead to the induction of a mature, glucose-sensitive phenotype on the dedifferentiated β-cells that appear during T2D progression [124]. Moreover, the importance of HMG20A in both neuronal development and islet β-cell maturation pinpoints this factor as a possible common regulator of pancreatic and CNS-mediated glucose homeostasis [120]. However, further studies are required to validate the possibility of targeting HMG20A or its downstream effectors as a novel regenerative therapy for diabetes aiming at the redifferentiation of the dedifferentiated β-cells.

## 5. Concluding Remarks

Controlled manipulation of a functional β-cell mass in humans, meaning both the replacement of the lost β-cell mass and the restoration of β-cell functionality, represents a holy grail for therapeutic interventions in diabetes. The possibility of stimulating the intrinsic plasticity of adult islets under prediabetic or diabetic conditions for the regeneration of the lost functional β-cell mass could become a successful curative treatment for this chronic disease of pandemic proportions worldwide. However, further studies are still needed to demonstrate the feasibility of this therapeutic approach.

Different studies using mouse models, combined with powerful high-throughput screening technology employed to characterize islet cell populations in both rodents and humans, have allowed the identification of pathways/mechanisms involved in β-cell heterogeneity. In addition, several studies have shown the high relevance of the maintenance of the different β-cell subpopulations for the islet functionality, since alterations in the proportions of different β-cell populations is sufficient to alter insulin secretion in response to glucose. However, the experimental differences among these studies, using different animal models and/or selecting different maturation reporter genes for the analysis of the islet populations, have led to the description of numerous subpopulations of β-cell that may overlap, at least to some extent. Big metadata analysis of the different identified subpopulations in both humans and rodents is mandatory to fully understand the heterogeneity of the β-cells and the genetic and epigenetic differences between them. Moreover, these studies will allow the identification of novel surface markers required for the identification and isolation of live β-cell subpopulations necessary to translate the data derived from rodent models to human islets. These meta-analyses will also help in the identification of the molecular mechanisms driving the loss of β-cell identity in the dedifferentiated β-cells, which will likely reveal novel druggable pathways for avoiding/reversing this dedifferentiation of β-cells. Moreover, as evidenced by the discrepancies in the GABA effect on ⍺- to β-cell transdifferentiation, combined efforts of different groups will provide a clearer picture of the involvement of the different mechanisms described for β-cell neogenesis in the regeneration of the lost β-cell mass under different metabolic challenges.

## Figures and Tables

**Figure 1 ijms-22-04239-f001:**
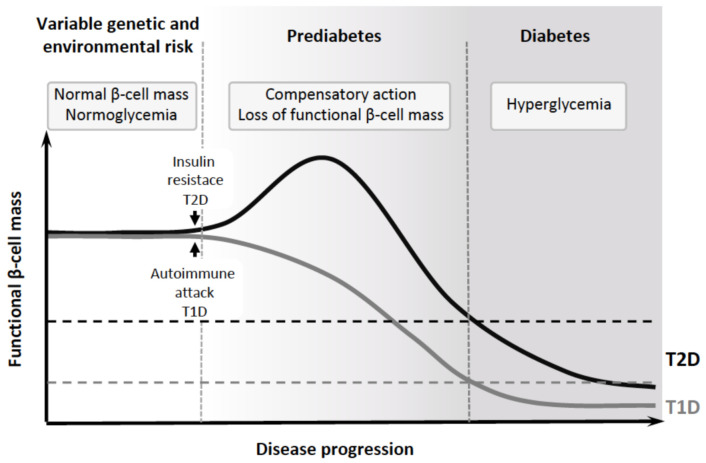
Diabetes progression stages. Diabetes is a continuum that is not clinically symptomatic until the levels of functional β-cell mass are below a threshold (dash line) that disables the capacity of the islets to compensate for the insulin demand. However, the alterations in the functional β-cell mass appear at earlier stages, before the onset of hyperglycemia.

**Figure 2 ijms-22-04239-f002:**
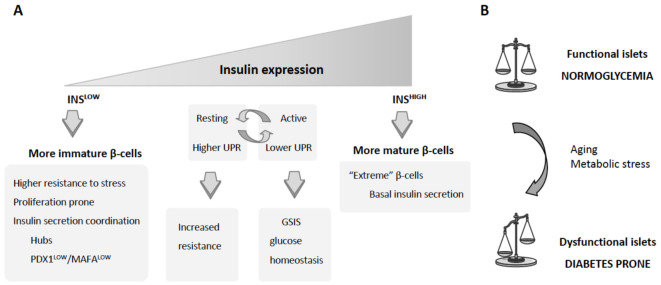
Heterogeneity of islet β-cells. (**A**) Differences in insulin and other β-cell marker genes give rise to the apparition of different β-cell subpopulations with different functionalities. (**B**) The maintenance of the proportions of these subpopulations ensures the adequate function of the islets. Aging and stress conditions can alter these subpopulations with detrimental consequences for islet functionality and adaptation that can eventually lead to the development of diabetes under stress conditions.

**Figure 3 ijms-22-04239-f003:**
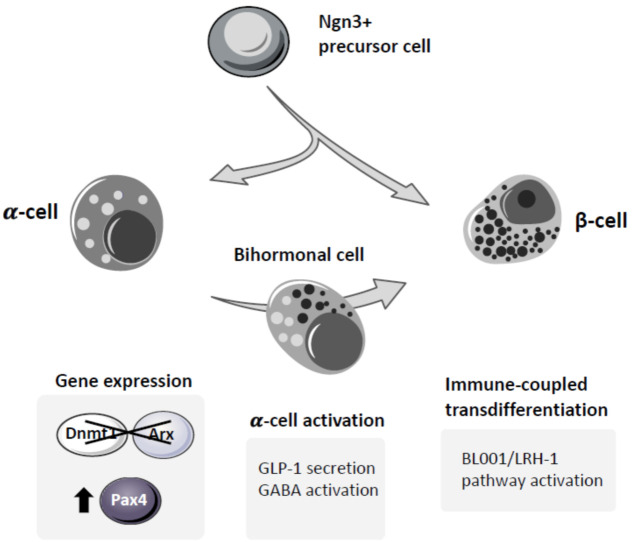
⍺- to β-cell transdifferentiation. The common origin of ⍺- and β-cells resulted in permissive epigenetic marks in ⍺-cells that under specific circumstances can activate their transdifferentiation into β-cells. This reprogramming of ⍺-cells into β-cells that is activated in islets under stress conditions is not able to regenerate an adequate β-cell mass to recover the endogenous control of glycemia. However, this process that involves a close crosstalk between islet cells as well as with the immune system cells can be stimulated by different treatments, such as GLP-1 and likely GABA, as well as the novel small molecule BL001.

**Figure 4 ijms-22-04239-f004:**
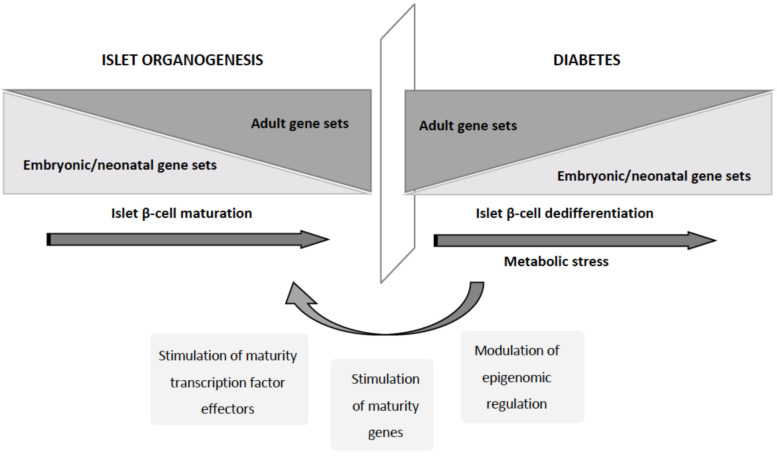
Redifferentiation of islet β-cells. The metabolic stress-induced overload of β-cells triggers their dedifferentiation in an attempt to avoid cell death. This process is due to the reactivation of gene set characteristics of the immature β-cells. Nevertheless, this process can be reversed since the dedifferentiated β-cells retain the potential to be redifferentiated.

## Data Availability

Not applicable.

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
