# Peer review of "Harnessing the Endogenous Plasticity of Pancreatic Islets: A Feasible Regenerative Medicine Therapy for Diabetes?"

_ijms, 2021, doi:10.3390/ijms22084239_

Round 1

Reviewer 1 Report

Lorenzo and colleagues have written a comprehensive review on possible mechanisms for increasing functional beta-cell mass with the aim of regaining glucose homeostasis. Three main strategies are considered: (i) exploiting the heterogeneity of beta-cells where certain subpopulations may be more susceptible or resistant to environmental stressors, (ii) transdifferentiation of other cell types within the islet to beta-cells, and (iii) promoting redifferentiation of beta-cells.

The manuscript is well written with just a small number of typographical errors noted by this reviewer (outlined below). As a minor point, the authors could consider including sub-headings as the large slabs of text were sometimes difficult to get through without distraction. I would also reconsider the use of the word “palliative” with reference to insulin therapy. This is a highly emotive term could be considered stigmatizing. 

Line 15: “led to a search for novel…”
Line 25: something appears to have gone wrong with this sentence and/or the formatting?
Line 41: use of “however” is unnecessary
Line 54: suggest replacing “not detected” with “not clinically symptomatic” since islet autoantibodies are detectible prior to significant loss of functional beta-cell mass
Line 91: “Evidences” should be “Evidence”
Line 153: remove comma after “hubs,”
Line 370: include a space between “promoting” and alpha
Line 479: “dedifferentiates” should be “dedifferentiated”
Line 486: add closed bracket after reference [96].
Line 505: remove bracket in front of “Modulating” 

Reviewer 2 Report

This review by Lorenzo, Gauthier and coworkers provides a comprehensive summary of the current understanding on the islet beta cell heterogeneity and plasticity, and discusses several scenarios of exploiting these phenomena to restore functional beta cell mass for treating diabetes. The article was well written and represents a timely addition to this important topic.  Comments below are intended to enhance this manuscript.

The authors address extensively on the beta cell heterogeneity, a topic that has been under intense investigation in recent years. Heterogeneity of other islet cell types including alpha cells (for example, Pubmed ID  32668245,  21078586 etc) is an emerging topic.  Since the manuscript elaborates on alpha-beta cell trans-differentiation,  mentioning or expanding on alpha cell heterogeneity would be appropriate.

Thus far the majority of studies on beta cell heterogeneity has been carried out in mice, and little is known about their relevance to human beta cells.  As mentioned in this review, the study by Grompe and coworkers used CD9 and ST8SIA1 to distinct human beta cells into 4 groups. However, this grouping turned out to be rather inefficient, and there has not been any follow up study on this subject since its publication in 2016. To facilitate detailed molecular, cellular, and functional analysis of human beta cell subsets, the field desperately needs cell surface markers or dye labels to distinguish and to isolate subpopulations of LIVE cells. The authors may consider expanding this point. This roadblock also extends to mouse islets (albeit to a less extent), where we are limited by the genetic reporters.     

Page 13, top : “Controlled manipulation of β-cell mass in humans represents a holy grail for therapeutic interventions in diabetes.”  This may represent a different opinion - but restoring beta cell function is more likely the holy grail, at least for T2D (Pubmed 32333873)

 Minor:

Abstract  - Parts of the last sentence were missing ?
